# A Design Contribution to Ergonomic PC Mice Development

**DOI:** 10.3390/ijerph19138126

**Published:** 2022-07-02

**Authors:** Miguel L. Lourenço, Rui A. Pitarma, Denis A. Coelho

**Affiliations:** 1Technical Scientific Unit of Engineering and Technology, Research Unit for Inland Development, Instituto Politécnico da Guarda, 6300-559 Guarda, Portugal; mlopes@ipg.pt (M.L.L.); rpitarma@ipg.pt (R.A.P.); 2CMAST—Centre for Mechanical and Aerospace Science and Technology, Universidade da Beira Interior, 6201-001 Covilhã, Portugal; 3CISE—Electromechatronic Systems Research Centre, Universidade da Beira Interior, 6201-001 Covilhã, Portugal; 4Supply Chain and Operations Management Department, School of Engineering, Jönköping University, 55111 Jönköping, Sweden

**Keywords:** ergonomic product development, ergonomic computer mouse design, human-systems interaction, usability

## Abstract

Developing new manual computer pointing devices abiding to the requirements set out in ergonomic product design literature necessitates joining contributions from several areas, including the ergonomic guidelines applicable to hand tool design, human–system interaction, and certain user characteristics such as anthropometric data. Computer mice are hand tools enabling the interaction with the computer, for use by people from both sexes and practically all ages. Moreover, the PC mouse’s intensive usage is able to cause musculoskeletal disorders. This paper reports on a study aimed at developing new computer mouse shapes, reducing forearm pronation, and providing hand–palm holding, supported by a literature review and an adequate design methodology, starting from known shapes of commercial products, the traditional (horizontal) computer mouse, and the vertical computer mouse. In this regard, potential concepts were generated as solutions to the previously specified problem through a set of creative tasks based on the specifications. Four new shapes were proposed to be evaluated through an assessment matrix; as a result, two new PC mice geometries were designed and fully prototyped. This study also reports on selected results of usability and an electromyographic evaluation of the prototypes against three commercial PC mice (horizontal, slanted, and vertical) by a sample of 20 participants, supporting validation of the development process and the newly developed geometries, with emphasis on the slanted conical innovative shape.

## 1. Introduction

Computer usage can be associated with the development of upper extremity pain; particularly, intensive mouse use can lead to hand and forearm musculoskeletal pain/injuries [1]. On the other hand, the extended use of the PC mouse is bound to endure, because in precision computer tasks such as dragging and steering graphical targets, continuously needed in some computer applications, such as CAD, touch screens have, so far, not been able to replace the mouse [2]. Modern computer users use the mouse almost three times as much as the keyboard [3]. In this context, using the traditional horizontal mouse leads to postures that increase the risk for injury. Therefore, it is desirable to improve upper extremity posture while using a computer mouse. Regarding this problem, previous studies have found posture benefits associated with using alternative mouse designs, and have concluded that increasing mouse height and slanting the mouse’s top face can improve wrist posture without negatively affecting performance [3]. In this line of thought, Lourenço et al. (2017) [4], in an experimental set-up with 20 participants, performed usability evaluation comparing a standard mouse (*Microsoft Optical 200*) with an alternative vertical mouse (*Evoluent VerticalMouse 4 Right*), supporting the adoption of a neutral pronation forearm posture. The results of the reported comparison suggest designing hybrid configurations of computer mice to achieve a compromise between the usability parameters and the expected long-term effects on health. Previous studies suggested that a proper mouse weight could benefit users in terms of increasing movement efficiency; its dimensions and geometry should be based on anthropometry, hand gestures, and comfortable hand postures [5]. Additionally, the weight of the mouse seems to affect forearm muscle activity during speedy operation, supporting the use of lighter devices [6]. Another factor that seems to make a difference during mouse usage is the hand size of the subjects, affecting grasp position and the level of muscle activity, suggesting that a computer mouse ought to be chosen according to the size of the subject’s hand [7]. Moreover, previous tests performed on a conventional (horizontal) PC mouse revealed a statistically significant association between hand width and the effectiveness of dragging graphical targets with the middle (scroll) button of the mouse [8]. Another study carried out a set of usability evaluation tests of similar flat shape PC mice, involving 30 undergraduates. They used their own and other subjects’ PC mice. Hypothetically, subjects would not experience improved efficiency when switching to other devices from their own device. The other devices shared the archetype of the owned device, but differed in dimensions and shape details, or activation thresholds. Based on a literature review and prior experimental results, the authors suggest tentative explanations to support the understanding of cases of subjects’ improvement in efficiency when changing to unfamiliar pointing devices. Variables such as the contour and fingers’ support and the surface finish of the device are suggested as relevant, together with the relative size of the device in relation to the size of the hand, corroborating previous studies [9]. In this sense, the authors propose alternative hybrid shapes supported by a structured product design method, resulting in the development, prototyping, and implementation of an alternative slanted PC mouse [10]. Hence, the present study was conducted by compiling a set of requirements, recommendations, and guidelines towards improvements in respect to characteristics previously identified in the literature, focusing on ergonomic and usability considerations, proposing a method to develop ergonomic computer mice. Thereby, four PC mouse shapes (variants) were proposed; full-scale mockups were made based on which to apply a suitable assessment matrix; and, as a result, two fully functional computer mice models were selected (preferred) and prototyped.

## 2. Materials and Methods

### 2.1. Operational Model for Computer Mice Geometry Development

A bespoke design method was developed and applied to support attaining the goals set forth for the present study (Figure 1). A task clarification was accomplished through the activities of product goals definition (objectives), analysis and definition of product requirements, and the setting up of the product specifications with support from a literature review. Concept generation was attained by means of sketches and, mainly, the execution of mockups, leading to a structured presentation of alternative concepts. The evaluation and refinement stage involved the definition of an evaluation (decision) matrix derived from the specification, leading to the selection of the best concepts following the previously implemented criteria, and improvement and refinement of the physical mockups (embodying specific characteristics from specification stage). Four models (variants) were proposed based on the specifications, and full-scale mockups were made to apply the assessment matrix. As a result, two fully functional models were developed and prototyped. The detailed design stage was conducted in view of prototyping, as well as ergonomic assessment and usability evaluation. The generic operational model for product design proposed by Hales [11], reviewed by Lewis and Bonolo [12], and cited by Coelho [13] was adopted as a reference. In an industrial design context, the subordinate processes are divided into five phases: (1) task clarification; (2) concept generation; (3) evaluation and refinement (of concepts); (4) detailed design (of the preferred concept); and (5) communication of results (through the refined prototype). However, the main goal throughout this design process was to attain appropriate physical prototypes for comparative ergonomic evaluation (validation). Hence, some adaptations were undertaken from the reference operational model (Figure 1). In this regard, subordinate processes 3 (evaluation and refinement) and 4 (detail design) were grouped. Moreover, the study on the production processes is not reported in this document, as well as subordinate process 5 (communication of results), as it is assumed to be out of context within the scope of this manuscript, focusing on the design with the goal of performing the product evaluation emphasizing the detailed ergonomic study. Thus, the development of functional prototypes occurred upstream of the communication of results, since these were indispensable to the ergonomic study, which included studies of usability and muscle activity of the participants.

In Figure 1, the dashed lines show the stages/operations that were not formally carried out according to the operational model of the design process, regarding the specificity of the current study. Detailed drawings were needed as early as in the process of developing the models. After refinement of the models, it was necessary, again, to use detailed designs, which are fundamental in the process of executing functional prototypes. Shapes, dimensions, and the mouse buttons’ positions were tested during the upgrade and refinement of the mockups. The detailed drawings integrated the tridimensional computer-aided design necessary for the 3D printing process adopted in the production of both the physical mockups and the functional prototypes. The prototypes were developed for comparative tests with commercial models. The flow of information shown in Figure 1 can occur between the several stages, with the possibility of returning to previous steps from downstream stages. This iterative process allows improving and refining the models. Some requirements emanating from the task clarification stage could only be applied on the functional prototype, including those dependent on the mechanisms and electronic circuits housed inside the prototype. For example, the force required to activate the buttons is only possible to measure on the fully functional prototype.

#### 2.1.1. Development of Task Clarification

The new models (geometry) should embody features fitting the requirements of the applicable standards, the principles of ergonomics for hand tools and work with a computer, and the requirements resulting from the analysis of the specific scientific literature. Intermediate slopes between 90° and 0° are desired. The use of a vertical mouse (90°) leads the forearm to assume neutral posture (0° pronation). On the other hand, the use of a flat mouse (0°) leads the forearm of the user to assume a full pronation posture. The prototypes should use the same hardware as a selected reference computer mouse to study (validate) the geometry compared to other models (geometries) available in the market. Thus, the hardware of the *Microsoft Optical Mouse 200* (mechanisms and electronic circuits housed inside) was implemented inside the new model prototypes.

The requirements and recommendations for computer mice design were collected from standard ISO 9241 (Ergonomics of human-system interaction) [14,15,16,17], as well from scientific publications [6,18,19,20,21,22,23,24]. The specification listed below was developed based on these requirements and recommendations. The design specification items are listed in Table 1.

#### 2.1.2. Concept Generation

In the conceptual stage, potential concepts were generated as solutions to the previously specified problem, through a set of creative tasks based on the specification presented in Table 1. Although this method may seem reductive from the point of view of the creative process inherent to the generation of concepts compared to the 2D sketch, it was possible to generate distinct 3D geometries. The four generated concepts are presented in Figure 2 as viewed from four alternative perspectives. In this stage, an attempt was made to incorporate the characteristics (requirements) necessary to fulfill the specifications, establishing compromises in the satisfaction of conflicting requirements, through shape solutions. Examples of requirements/recommendations that revealed to be conflicted were: providing grip surfaces of sufficient size and shape to prevent slipping, providing a support to the palm of the hand, providing finger anchoring to ease the movement of the pointing device with less effort of the fingers, promoting a close distance between distal ends of the index finger and thumb (precision grip), promoting postures in which the adjacent fingers (middle finger, ring finger, and pinky) do not assume different positions from each other, creating innovative aesthetic shapes, among others. The new shapes were manually generated using modeling technics with clay [25]. This material allows for easy manual modeling when slightly heated. Despite the lack of mechanical and electronic functions related with the use of the (nonexistent) functional buttons, the use of these primary models enabled the accomplishment of preliminary tests, related to form and functional movement of the solid object. Several aspects related to the postures of the hand fingers, hand–palm support, and anchoring could be tested. It was thus possible to generate new refined solid geometries, adapted to the anatomy of the hand, despite the difficulties that this task imposed [17] due to the differences between men and women’s anatomy and anthropometry.

Figure 2 shows the four concepts generated. The *pg* concept privileges the hand prehensile action and precision grip, in which the head of the object is manipulated between the tips (pads or sides) of the fingers and thumb. In view of the other generated geometries, the *pg* geometry implies greater difficulty in integrating the electronic and mechanical devices needed for the tests to validate the new geometries. The base surface of this geometry does not allow as effective a contact with the work surface (desk) as the other proposed geometries; this may compromise balance during dynamic use. The space available for inserting buttons is limited. The *pt* concept privileges the palm’s rest and wrist’s rest; although the resulting slanted small angle does not prevent the forearm’s pronation. In view of the other geometries, the recessed wrist support increases the contact surface with the work surface, and can also increase the effort required during its dynamic use (handling), further impairing the accommodation of different-sized hands. The *ch* concept emphasizes palm–hand rest, keeping the fingers slightly flexed (gently curved), getting a gently slanted posture of the wrist, while minimizing the forearm pronation. Compared to the other geometries, it has a less innovative shape, although with an inclination of around 30° and an unusual recess that provides an effective support for the thumb. Finally, the *ci* concept emphasizes the slanted posture of the wrist (and forearm) and the hand prehensile action. The *ci* geometry, compared to the other geometries, is one of the most innovative, allowing to reconcile a greater number of requirements according to Table 1. For example, it allows the adoption of a more curved posture of the fingers, and even a slightly recessed support for the wrist; it also seems to facilitate the accommodation of different hand dimensions in relation to the other proposed slanted geometries. The *pg* and *ci* concepts pursue the precise manipulation of an object located between the index finger, middle finger, and thumb, similarly to handwriting and other accurate hand operations. The latter condition is exceedingly difficult to achieve in the *pt* and *ch* concepts, because for these models, the precision manipulation is too dependent on the hand length of the user.

#### 2.1.3. Evaluation, Refinement, and Detailed Design of Preferred Concepts

From the specification and subsequent four concepts, an evaluation matrix composed of 16 criteria was defined. The decision matrix is presented in Table 2. Each assessment criterion was previously assigned a weight ranging from 1 to 3. The value 1 was attributed to characteristics considered less important, and the value 3 was attributed to characteristics considered more important, in the present context. For each concept, a score ranging from 1 to 4 was attributed to the satisfaction of each of the criteria, the value 4 corresponding to the better classification. The evaluation matrix was then completed by multiplying the score given to each concept by the weight previously assigned to the respective criterion under consideration. The total score obtained was 83 points for the *pg* concept, 83 points for the *pt* concept, 102 points for the *ch* concept, and 104 points for the *ci* concept. The *ch* and *ci* concepts were then selected and advanced to the model-making process. In the process of the selection of the preferred concepts, the manually-produced clay models proved to be decisive for a thorough evaluation.

The improvement and refinement of the models corresponds to an iterative process to incorporate the desired characteristics. The 3D printed mockups were obtained through a reverse engineering (using a 3D scanner) process complemented with the transposition of the physical dimensions of the clay models by manual collection of the dimensions. Figure 3 illustrates the 3D-printed model of the *ch* concept. The *ch* geometry allows an effective curved support for the hand, and the reduction of pronation of the forearm, presenting 30° of inclination (Figure 3a). This geometry also has a pronounced cavity for the support of the thumb, allowing anchoring and stabilization during the device’s displacement. Figure 4 illustrates the 3D-printed model of the *ci* concept which offers support for the hand with 45° of inclination (Figure 4, Figure 5b and Figure 6b), reducing the pronation of the forearm even more, allowing a greater curved support for the hand, and promoting a close distance between distal ends of the index finger and thumb (precision grip). 

The 3D-printed mockups (Figure 3, Figure 4 and Figure 5) enabled testing the ease of movement of the device and the deviation of the fingers during the referred movement. These mockups also enabled testing the influence of the design of the buttons on the positioning of the fingers and the ease of actuating on the buttons, as well as the suitability of the shape and size of the surfaces in contact with the hand. The handling tests enabled refining the models, leading to the development of the functional prototypes. During the refinement of these working mockups, the technical drawings needed to develop the functional prototype using 3D CAD techniques (Figure 6) were improved.

Figure 6 shows the digital models (*ch* and *ci*) during their three-dimensional parametric modeling, and the slanted angles measured from the front view. Figure 7 shows the shape and size of the reference model (*Microsoft Optical Mouse 200*), the mechanical components and electronic circuits of which were implemented in both model prototypes *ch* and *ci* (Figure 8). The prototypes were materialized in two parts each (shell and bottom base cover) using ABS thermoplastic with the Fused Deposition Modeling technique. The iterative process (Figure 1) led to the final prototypes, followed by the remaining tests, such as the measurement of the force required to push the buttons (Figure 8), and the tests for validation of other ergonomic aspects of the new geometries. The previously referred process (3D scanning complemented with measuring the physical dimensions of the mockups to develop the functional prototype using 3D CAD) enables easily generating alternative sizes of the same geometry or varying the proportions of its main dimensions. 

### 2.2. Comparative Evaluation between Developed PC Mice and Benchmark PC Mice

#### 2.2.1. Graphical Test Tasks

For the test tasks to evaluate the PC mice geometries, the current study follows the structure used by Odell and Jonhson [3]. Thus, to test and compare the PC mouse models of interest (Figure 9), the graphical test tasks occurred in the following way: pointing large, medium, and small (Figure 10). These tasks present three levels of difficulty through the dimensions of the circular targets and overall diameter of the test task. Pointing standardized tasks were performed with a purpose-built software; errors and times to complete tasks were also accounted for and recorded by the same software. The elapsed time between the targets and time required to complete each task was counted for a fixed number of targets, starting with the mouse click on a central target (circle), and ending when the fixed number of targets was completed. All tasks were performed in two cycles. The pointing (and clicking) task included the random activation of 12 targets per cycle (of 18 possible), totaling 24 targets (Figure 10).

#### 2.2.2. Characterization of the Sample of Participants

To test the PC mice geometries, 20 young adult subjects were recruited (10 male and 10 female). All 20 participants were considered CAD practitioners because they had two or more years of CAD training and practice, and all of them were right-handed and had normal or corrected-to-normal vision. Table 3 shows sample data.

#### 2.2.3. Efficiency Calculation

The efficiency in pointing (and clicking) using each mouse geometry during the tests was calculated according to the equations detailed in a previous study [27]. The effectiveness of pointing was calculated according to Equation (1), and the efficiency was calculated according to Equation (2).
(1)$${\mathrm{efa}}_{}=1-\frac{\mathrm{No}.\;\mathrm{FailedTargets}}{\mathrm{No}.\mathrm{TotalTargets}}$$
(2)$${\mathrm{efi}}_{}=\mathrm{efa}\times \frac{\mathrm{minimum}\;\mathrm{mean}\;\mathrm{completion}\;\mathrm{TIME}}{\mathrm{mean}\;\mathrm{completion}\;\mathrm{TIME}\;\left(\mathrm{subject}\right)}$$

efa—effectiveness of pointing (and clicking).efi—efficiency of pointing (and clicking).No. FailedTargets—number of failed targets by the subject.No. TotalTargets—total number of targets to be hit.minimum mean completion TIME—lowest mean completion time across the whole set of replications of participant–device combinations.mean completion TIME (subject)—mean time to complete the task for the participant–device combination.

#### 2.2.4. Muscular Activity Assessment

The activity of a set of forearm muscles was assessed through surface electromyographic sensors (S-EMG) during the performance of the pointing (and clicking) task with each of the mice geometries. Thus, *Extensor Digitorum Communis* (ED), *Extensor Carpi Ulnaris* (ECU), *Extensor Carpi Radialis* (ECR), and *Abductor Pollicis Longus* (APL) muscles, the activity of which is commonly recognized as related with computer mouse usage, were selected. ECR activation is more related with wrist extension and radial deviation, ECU with ulnar deviation of the wrist, ED with finger extension, and APL with abduction of the thumb. More details about muscular activity assessment related with the current study can be found in Coelho and Lourenço [28]. EMG data were normalized with maximum voluntary contraction (MVC) values for each muscle (ED, ECU, ECR, and APL), and, for each subject, were then transformed to obtain the amplitude probability distribution function (APDF) of the electromyographic signal [29]. The probability of amplitude at a certain level of muscle contraction is the probability of myoelectric activity being less than or equal to that level of contraction, and may be expressed as the fraction of the total duration at which the signal is less than or equal to that level. APDF10 has been recognized as related with baseline activity, APDF50 as median activity level, and APDF90 as peak activity [30].

## 3. Results and Analysis

IBM SPSS was used for statistical analysis of the data. Figure 11 shows the mean efficiency in the pointing task (pointing large, medium, and small) by mouse geometries. A RM-ANOVA with Bonferroni correction was conducted to investigate the impact of PC mouse geometry on the mean efficiency achieved performing the tasks. There was a significant main effect of mice geometries (F(2.736; 51.976) = 16.155; *p* < 0.0005) on the efficiency in the pointing large task, as well in point medium (F(2.635; 50.072) = 14.995; *p* < 0.0005), and in pointing small (F(3.367; 63.979) = 27.164; *p* < 0.0005). Pairwise comparisons with Bonferroni adjustment for multiple comparisons revealed several significant differences between the different geometries in all three degrees of pointing tasks (Figure 11). The developed *ch* geometry revealed better efficiency compared with the *ev* vertical benchmark mouse in all levels of the pointing task: pointing large *ch*–*ev* (*p* = 0.001), medium *ch*–*ev* (*p* = 0.001), and small *ch*–*ev* (*p* < 0.0005). On the other hand, the developed *ci* geometry revealed worse efficiency compared with developed *ch* geometry in pointing large *ch*–*ci* (*p* < 0.0005) and in pointing medium *ch*–*ci* (*p* = 0.004). From the point of view of pointing task efficiency, the overall results show that the new developed geometries stand between the *ev* and *mi* benchmark mice, and it seems that the newly developed *ch* geometry is very close to the *ak* benchmark mouse.

A muscle-by-muscle overview of the results of APDF10, APDF50, and APDF90 by PC mouse geometry, obtained from the pointing large graphical task, is shown in Figure 12. A RM-ANOVA, with Bonferroni correction for APDF50, was conducted to investigate the impact of PC mouse geometry on the activation of each monitored muscle (Figure 12). There was a significant main effect of mice geometries (F(2.585; 49.112) = 4.352; *p* = 0.012) on APL muscle activation, although pairwise comparisons with Bonferroni adjustment for multiple comparisons did not reveal significant differences between the different geometries.

## 4. Discussion

The common PC mouse user may think that it does not make any difference to use one or another mouse, and that all the available devices are scientifically supported. There is a great variety of computer mice available on the market, and there seems to be no scientific support for all these geometries. As is known, there are multiple requirements imposed on computer pointing devices, necessitating establishing trade-offs between seemingly conflicting requirements, which implies that a large variety of alternative designs may be created from the same product specifications. The present study is targeted towards satisfying a set of requirements, recommendations, and guidelines, towards improvements in respect to characteristics previously identified in the literature as requiring improvement. Usability evaluation and ergonomic assessment were deployed to sort out the complexity, and validate the design and development process presented against a set of alternative benchmark commercial models (including a horizontal, a slanted, and a vertical model). Table 4 presents the salient characteristics, from a favorable and unfavorable viewpoint, for each one of the five PC mouse geometries tested for usability and muscular activation (monitored through surface electromyography of four forearm muscles).

Usability results are comparable to the benchmark models; in particular, for the slanted and vertical models (*ak* and *ev*). Surface electromyographic study for the pointing task shows a significantly decreased APL activation for the *ci* model at APDF50, pointing to the interest in industrializing and marketing this geometry for enhanced computer mouse ergonomics. Future studies should consider testing the new geometries using an array of other kinds of tasks, as well as including gaming applications in the tests, and the test by computer games, where added speed is valued in addition to ease of use and comfortable handling. 

The current results are aligned with previous work by Odell and Johnson [3] suggesting that increasing mouse height and angling the mouse can improve wrist posture, without negatively affecting performance; this is particularly noticeable in prototype *ci*, but also noticeable for prototype *ch*. Moreover, the current study also corroborates previous work by Agarabi et al. [7] showing a significant decrease in the level of sEMG activity for selected muscles when subjects were tested using ergonomic computer mice, which, in the current study, was applicable to APL in pointing large for APDF50, with regard to the ci geometry. Similar to previous results of a study investigating the use of mice allowing a more neutral posture of the wrist [18], the current study shows that the *ci* prototype mouse falls well within the range of performance measures associated with already existing commercially available input devices.

## 5. Conclusions

A systematic product development methodology was deployed to create new geometries of computer mice through the requirements, guidelines, and recommendations emanated from a specialized literature review. The proposed methodology of the development of PC mice was surveyed and compiled from the ISO 9241 (Ergonomics of human-system interaction) standard series, and from applicable previously-published scientific studies. A product requirements specification consisting of both qualitative, as well as quantitative, product requirements was elicited. Four conceptual sketches were assessed against a 16-criteria evaluation matrix and scored using weighted criteria, yielding the preferred concepts for further development. Mockups were generated and subject to preliminary testing (shape and fitness to hand anatomy and anthropometric dimensions). As an outcome of the development process depicted, two fully functional prototypes were unveiled, and quantitative physical testing ensued. The prototypes generated were successfully tested for usability and muscular effort, and validation was attained with greater emphasis for geometry *ci*, which shows added benefits in decreased levels of muscular activation for APL. 

The compromise between usability and the long-term health of users (according to the prevention of musculoskeletal disorders incurred from using PC mice with a pronated forearm posture) is probably only fairly judged after a medium-to-long-term use of different PC mice geometries. Due to short-term assessments that do not factor in the long-term negative outcomes of geometries that lead to pronation, efficiency is favored in short-term evaluation. This paper contributes to promote heightened awareness of the need to establish a more balanced overview in a longer time, favoring sustained long-term improved user experience with healthy users. 

Following a systematic product development methodology, demonstrating an approach to principle-driven design, two new ergonomic computer mice were developed and prototyped to a fully functional state, enabling the subsequent validation of usability evaluation and ergonomic assessment. This study contributes to enhance the knowledge related to PC usage, specifically related to computer pointing device development, with focused contributions in the following aspects:compilation of requirements emanating from regulations and other related literature for specifications of new handheld computer pointing devices,method for developing and selecting (choosing) innovative PC mice geometries following previously defined criteria, andevaluation tools applicable to the products developed (usability, muscle activity).

## Figures and Tables

**Figure 1 ijerph-19-08126-f001:**
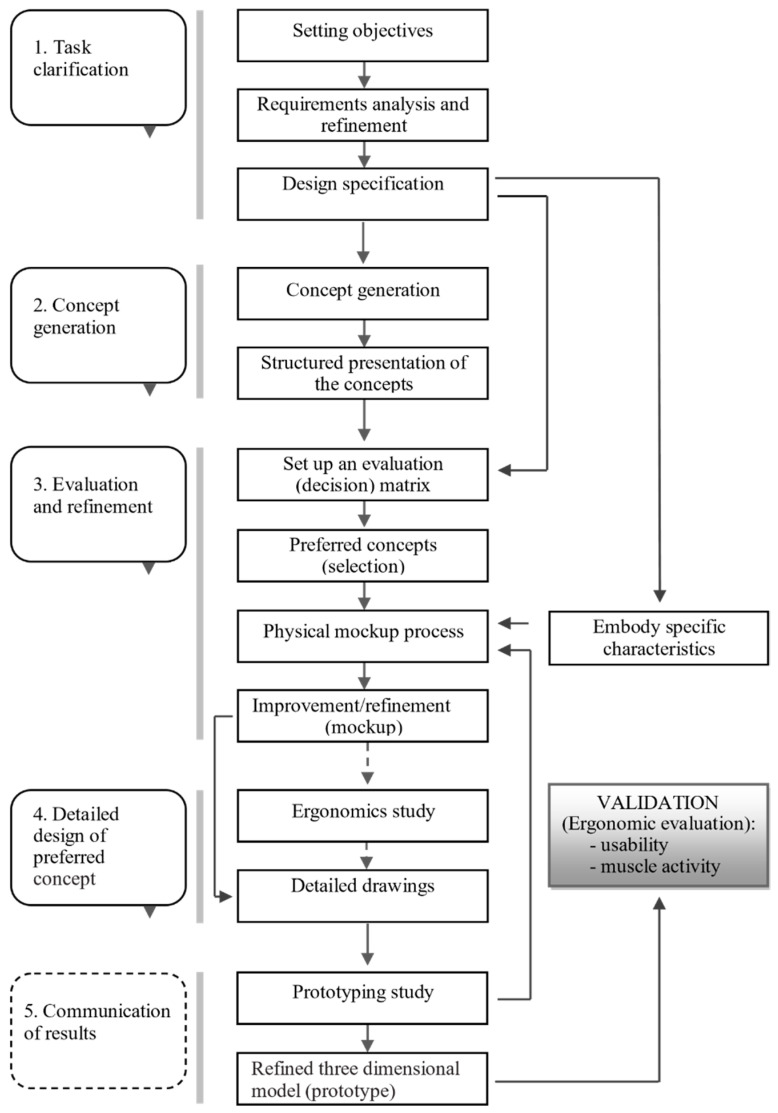
Process flow diagram of the operational model adopted in computer mice geometry development (adapted from Lewis and Bonollo [11], Hales [12]).

**Figure 2 ijerph-19-08126-f002:**
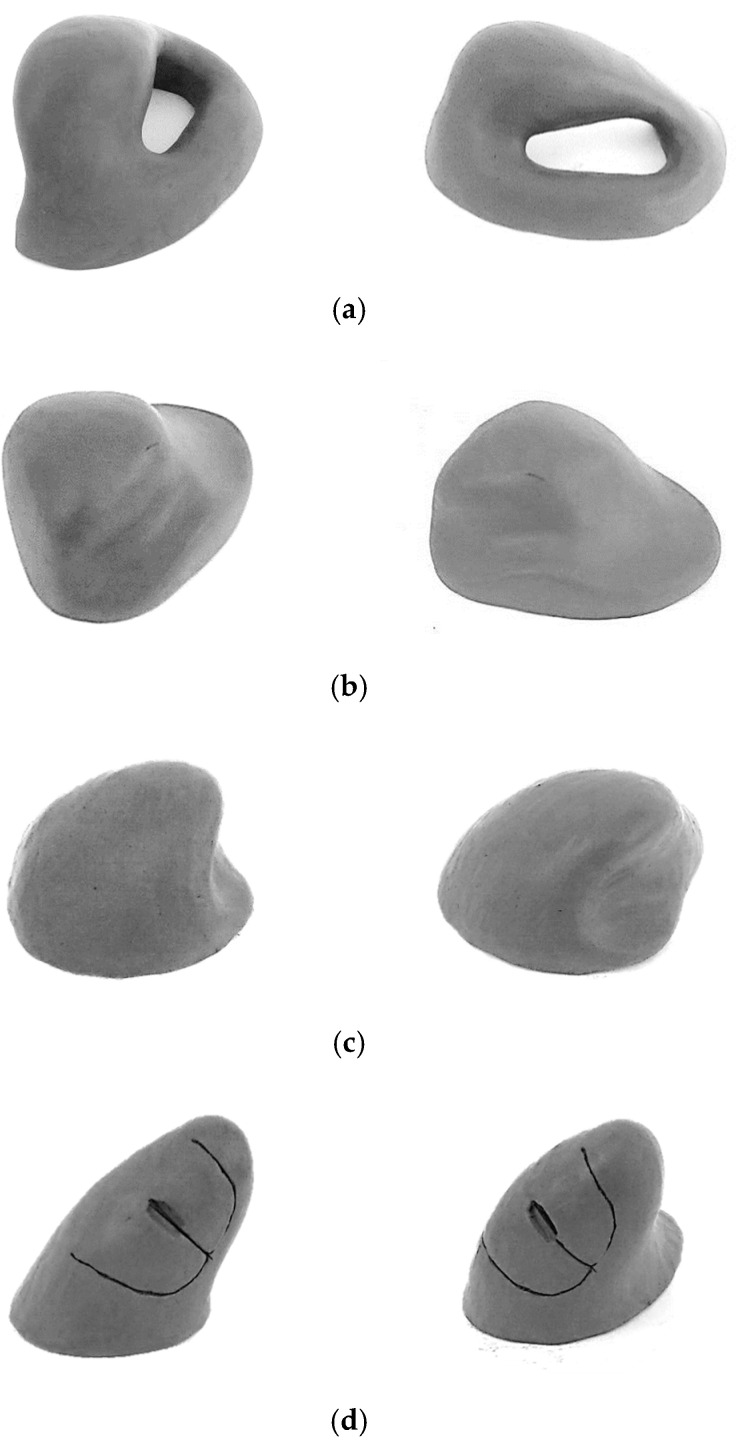
Concept generated. (**a**) *pg* concept; (**b**) *pt* concept; (**c**) *ch* concept; (**d**) *ci* concept.

**Figure 3 ijerph-19-08126-f003:**
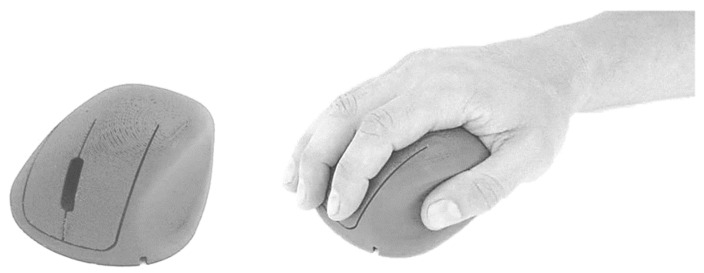
*ch* concept mockup (male adult hand, length 190 mm, width 88 mm, corresponding respectively to 40th and 50th percentile, according to Gordon et al. [26]).

**Figure 4 ijerph-19-08126-f004:**
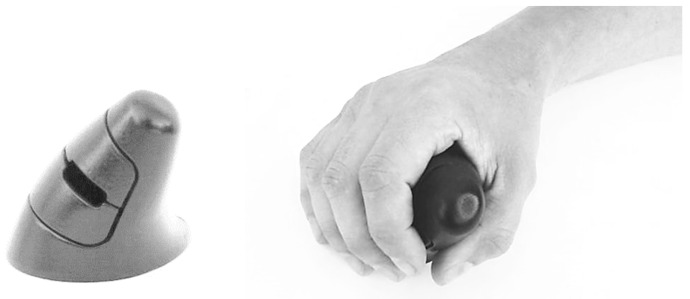
*ci* concept mockup (male adult hand, length 190 mm, width 88 mm, corresponding respectively to 40th and 50th percentile, according to Gordon et al. [26]).

**Figure 5 ijerph-19-08126-f005:**
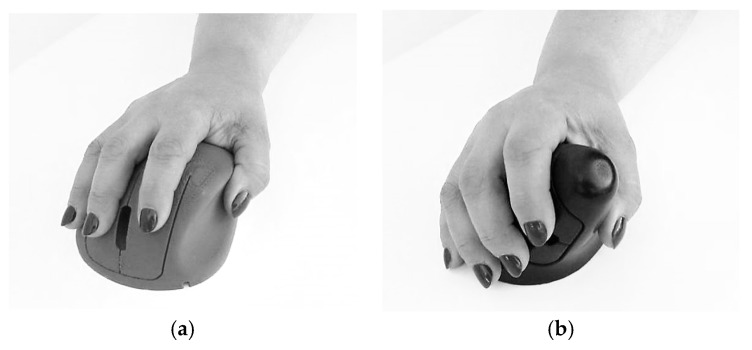
Concept mockups (female adult hand, length 165 mm, width 74 mm, corresponding respectively to 5th and 15th percentile, according to Gordon et al. [26]). (**a**) *ch* concept mockup; (**b**) *ci* concept mockup.

**Figure 6 ijerph-19-08126-f006:**
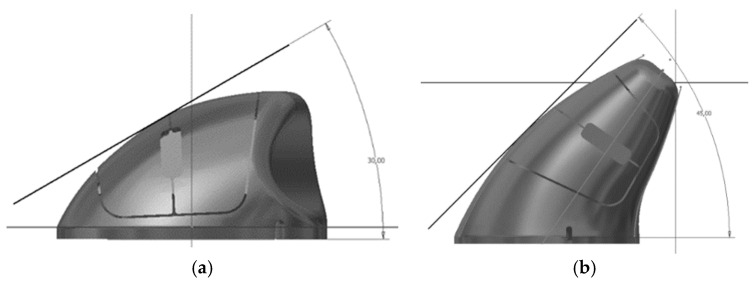
3D CAD models (front view). (**a**) *ch* concept (slanted angle of about 30°); (**b**) *ci* concept (slanted angle of about 45°).

**Figure 7 ijerph-19-08126-f007:**
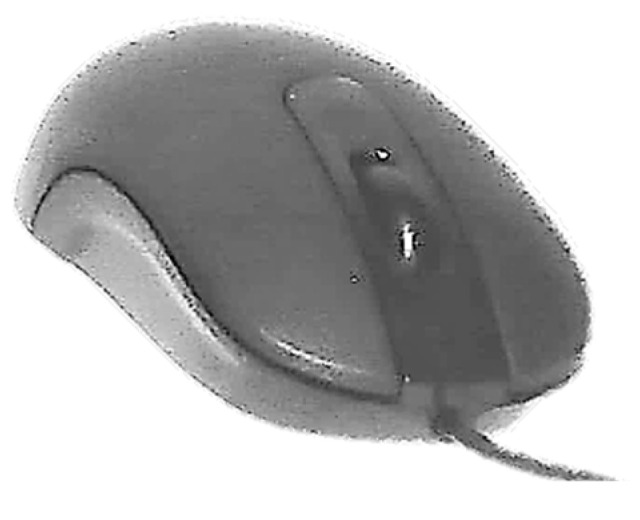
Device used as reference: *Microsoft Optical Mouse 200*.

**Figure 8 ijerph-19-08126-f008:**
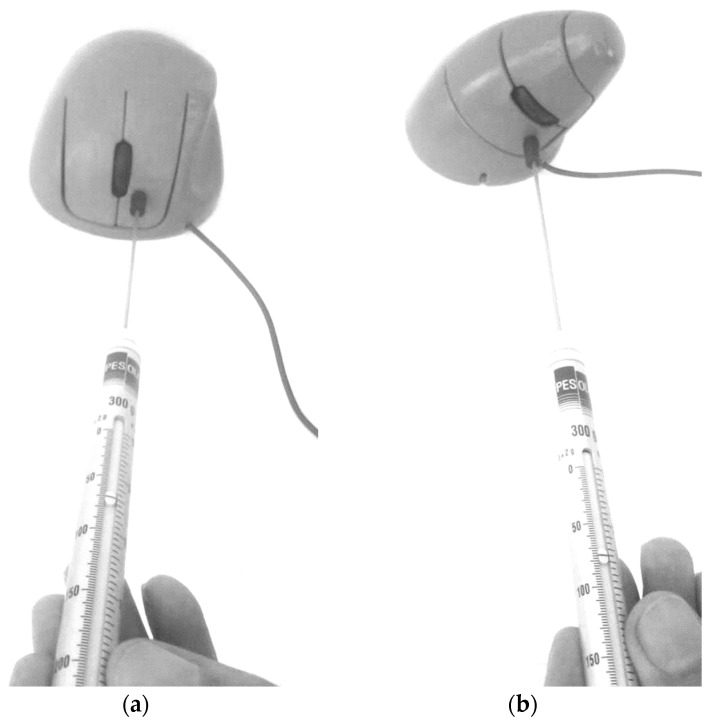
Measuring the force required to activate the buttons and dimensions of the PC mice’s fully functional prototypes. (**a**) *ch* prototype, force required to activate the buttons (70 gf = 0.687 N); (**b**) *ci* prototype, force required to activate the buttons (80 gf = 0.785 N).

**Figure 9 ijerph-19-08126-f009:**
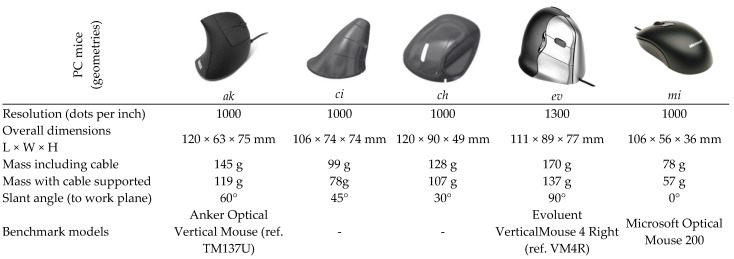
PC mice used in comparative evaluation through graphical tasks (ci and ch are the developed geometries).

**Figure 10 ijerph-19-08126-f010:**
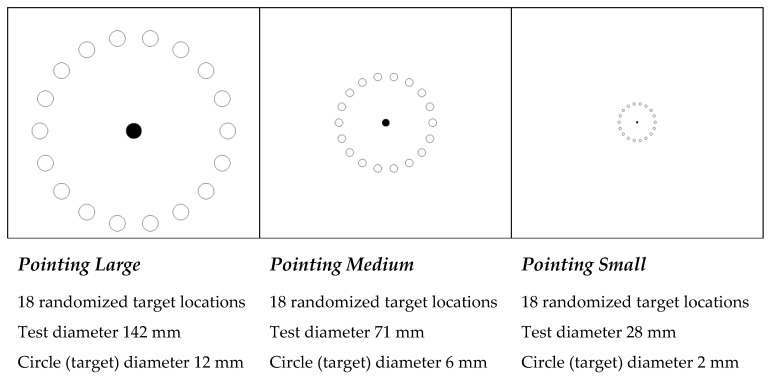
Pointing (and clicking) test tasks.

**Figure 11 ijerph-19-08126-f011:**
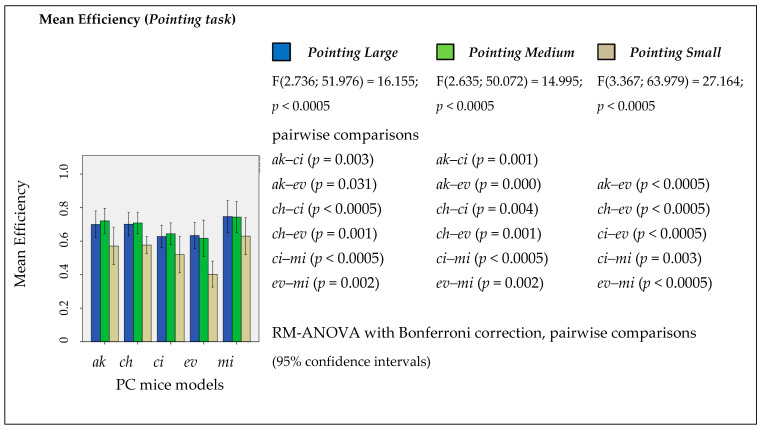
Mean efficiency by task evaluated between mice geometries with RM-ANOVA with a Bonferroni correction applied to pairwise comparisons.

**Figure 12 ijerph-19-08126-f012:**
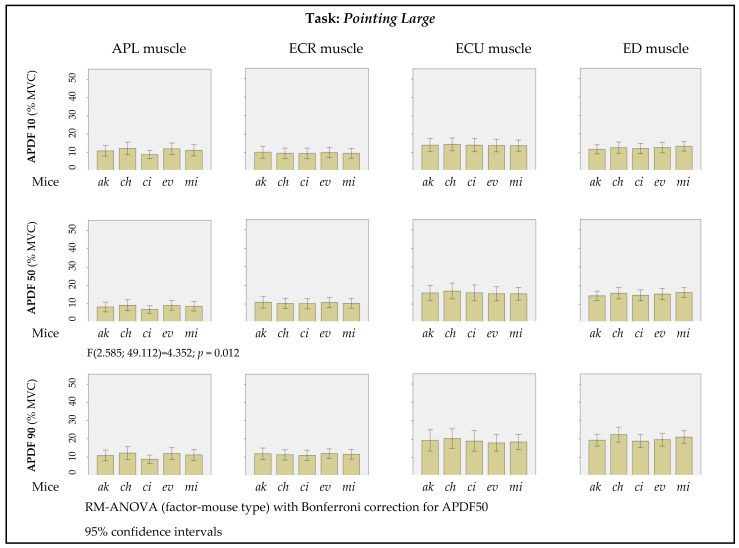
Muscular activity by PC mice geometry performing the pointing large task (APDF 10, 50, 90).

**Table 1 ijerph-19-08126-t001:** Design specification requirements.

Main Scope	Requirements and Recommendations
User posture-related and anthropometric-related requirements and recommendations	The device should be operated by the user without undue deviation of the hand, fingers, arm, shoulder, and head from their respective neutral positions.
	The device should be operated by the user without excessive effort; hence, the biomechanical load shall be minimized and the device shape shall take into consideration the minimizing of static muscle load.
	The device should minimize the need for extreme positions such as wrist extension, radial or ulnar deviation, and forearm pronation.
	The wrists and forearms should be near their neutral postures, avoiding wrist and finger extension; the most comfortable hand gestures are those where the wrists are kept straight, and the fingers are slightly flexed (gently curved) or in a loose fist.
	The device shape and the buttons’ locations should minimize finger extension or other movement or positioning that could cause finger strain or static load of the extensor muscles of any fingers.
	Input devices should be designed to accommodate the hand size of the intended user population.
Usability-related and innovation-related requirements and recommendations	The weight and inertia of the device should not degrade the accuracy during its use.
	The input device should be designed to be resistant to inadvertent button activation during its use, and it should be possible to press the buttons on the mouse without reducing control of the device.
	The device should promote an intuitive interface, adapting to skills already acquired, to minimize the learning threshold, and optimizing for perceived comfort.
	The input device should be effective, efficient, and satisfactory for the task being performed and the intended work environment.
	The intended use of an appropriately designed input device for a primitive task (such as pointing, selecting, and dragging) is either obvious or easily discovered.
	Buttons should be shaped to assist finger positioning and button actuation.
	Buttons should have a displacement force within the range of 0.5 N to 1.5 N until actuation, and should have a minimum displacement of 0.5 mm and maximum of 6 mm.
	The motion sensing point should be located under the fingers (precision grip posture) rather than under the palm of the hand.
	Grip surfaces should be of sufficient size, shape, and texture to prevent slipping.
	The device shall enable anchoring some part of the fingers, hand, wrist, or arm on it or on the worksurface, to create a stable relationship between the hand and the point of action.
	The new geometry should be innovative.

**Table 2 ijerph-19-08126-t002:** Evaluation matrix (weights for each factor ranging from 1 (less important) to 3 (most important); the rating for each factor ranging from 1 (worse) to 4 (better)).

Rating Criteria (Factor)	Weight	Concept							
		*pg*		*pt*		*ch*		*ci*	
		Score	Rating	Score	Rating	Score	Rating	Score	Rating
The use of the PC mouse shall enable anchoring some part of the fingers and/or the hand	1	4	4	2	2	2	2	3	3
The use of the PC mouse should minimize ulnar and radial deviation of the hand	3	3	9	1	3	2	6	3	9
The use of the PC mouse should minimize wrist extension and wrist flexion	3	4	12	2	6	2	6	3	9
The use of the PC mouse should minimize forearm pronation and forearm supination	3	1	3	2	6	3	9	4	12
The shape and location of the buttons should minimize finger extension and finger strain	3	4	12	4	12	4	12	4	12
The PC mouse’s shape should be designed to accommodate the hand size of the intended user population	2	2	4	1	2	2	4	3	6
The hand (fingers) should keep slightly flexed (gently curved) or in a loose fist when grasping the device	2	3	6	3	6	3	6	4	8
The PC mouse’s shape should avoid discordant adjacent fingers postures (middle finger, ring finger, and pinky)	2	3	6	4	8	4	8	3	6
The PC mouse’s shape and the buttons’ locations should avoid finger extension when clicking, or static load of the extensor muscles of any fingers	3	3	9	2	6	3	9	3	9
The PC mouse’s shape should facilitate the implementation of the most suitable buttons	3	1	3	3	9	4	12	2	6
The motion sensing point should be located under the fingers (precision grip posture)	2	1	2	4	8	4	8	3	6
The PC mouse’s center of gravity should be situated on the grasp axis regarding handle grasp	2	2	4	2	4	3	6	3	6
The PC mouse should adapt to skills already acquired	1	2	2	3	3	4	4	3	3
The physical characteristics of the PC mouse should conform to the established stereotypes	1	1	1	3	3	4	4	2	2
The PC mouse should promote an intuitive interface	1	2	2	3	3	4	4	3	3
The new PC mouse geometry should be innovative	1	4	4	2	2	2	2	4	4
Total weighted score		83		83		102		104	

**Table 3 ijerph-19-08126-t003:** Characterization of the sample of participants.

Participants			Age (Years)	
Number of Participants	Sex	CAD Practicioner	Mean (SD)	Range
10	Female	10	23.1 (2.7)	20–29
10	Male	10	25.4 (2.6)	22–30

**Table 4 ijerph-19-08126-t004:** Summary of relevant dichotomous highlights of the tested PC mice geometries.

Geometries (Slant Angle)	Favourable Salient Aspects	Unfavourable Salient Aspects
*ev* 90 deg	Neutral forearm posture (balanced between supination–pronation)	Lowest pointing and clicking efficiency (highest rate of errors in inexperienced use)
*ak* 60 deg	Best compromise between usability tests and electromyographic analysis performed	Shark fin geometry—not inclusive (restrictive hand sizes)
*ci* * 45 deg	APDF50 APL best of the tests (pointing large)	Buttons are hidden by the body of the PC mouse (medium error rate in inexperienced use)
*ch* * 30 deg	High pointing and clicking efficiency (low rate of errors in inexperienced use); thumb support included in the geometry	Low pointing and clicking efficiency (high rate of errors in inexperienced use)
*mi* 0 deg	Highest pointing and clicking efficiency (lowest rate of errors in inexperienced use)	Full pronation of the forearm

* These PC mice geometries (*ci* and *ch*) were designed and prototyped by the first author in 2017 for his PhD thesis; at that time, in addition to it not being possible to identify identic geometries from the scientific literature, there were no commercially available models embodying these PC mice geometries.

## Data Availability

The data presented in this study are available on request from the corresponding author. The data are not publicly available due to translation missing.

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
