# Peer review of "A Design Contribution to Ergonomic PC Mice Development"

_ijerph, 2022, doi:10.3390/ijerph19138126_

Round 1

Reviewer 1 Report

The iterative process used for the design of the mice looks useful and applicable in ergonomic designs, but the parameters used in the evaluation were a bit subjective. The results of the usability and muscular activity tests should have been included, which would allow validating how good the design was.

There are mice on the market that are very similar to those presented here, so it is not really possible to see a contribution to knowledge. A comparative evaluation between the designed mice and those on the market would have been interesting, as a way of demonstrating significant contributions.

Author Response

This article proposes a method to create/develop new PC mice geometries with reference to information emanating from regulations, ergonomic principles and related scientific literature; however, information was added regarding the evaluation and validation of the new geometries, whose content was intended to be used in future publications so as not to make this article too long and “heavy”.

Reviewer 2 Report

1.        Many typos need to be correct. Just for example in Ln 37-38. 

2.        Ln 131-133. It will be better to use a figure to express the concept.

3.        For finding the requirements, the method is too simple and not sufficient. Moreover, some requirements seem similar (See Table 1)

4.        For Figure 2, the types of the four concepts are been already easily found in the market. Lack of the novelty, and creation.

5.        More information for the four concepts are needed (e.g. Advantage, disadvantage, difference) 

6.        The innovation and improvement of the final two concepts need to address clearly. 

7.        Discussion is too less. To conduct an experiment to evaluate the human usability, physical responses and motions is highly recommended. Comparing the difference, improvement and advantage with the other available products are also needed. The discussion section has completely put the cart before the horse, including the objectives and implications, without seeing a meaningful discussion of the content of this study.

8.        The innovation point of the method proposed in this paper to evaluate the shape of pc mice products is not prominent enough.

9.        The introduction of the shape of mouse series products is not detailed.

10.     Figure1: ergonomics evaluation section is questionable. Firstly, please explain why the article uses usability and muscle activity, and their reliability. Secondly, the quantitative study of their factors is needed, which is what this paper lacks.

11.     The results section in lines 252-264 is incomprehensible and does not provide a good analysis and description of the data from table 2, as well the force and dimension measurements.

12.     Thirty-five and thirty-six lines said that computer users now use the mouse three times more than the keyboard, but this data is completely unsourced.

Author Response

  1. Many typos need to be correct. Just for example in Ln 37-38. Corrected!
  2. Ln 131-133. It will be better to use a figure to express the concept. Answer: The revision already introduces 3 new Figures and Tables and a lot of new content.
  3. 3. For finding the requirements, the method is too simple and not sufficient. Moreover, some requirements seem similar (See Table 1). Answer: In fact, this article contains a summary compilation of requirements emanating from regulations, ergonomic principles and scientific literature related to the development of manual pointing devices for computers, these requirements are sometimes somewhat ambiguous and in most cases no parameters are quantified , making it difficult to fully and objectively comply with these requirements. In Table 1, similar requirements were omitted or “merged”.
  4.  For Figure 2, the types of the four concepts are already easily found in the market. Lack of the novelty, and creation. Answer: from the consulted literature it was not possible to identify geometries exactly the same as the proposed models, whose respective description is presented throughout the manuscript, mainly in section 2.1.2. Concept Generation. As an example, “The ci geometry, compared to the Anker commercial model, allows the adoption of a more curved finger posture and a slight further back support for the wrist”, it also seems to facilitate the accommodation of different dimensions of the hand compared to the others. proposed inclined geometries, including the commercial Anker geometry. In addition to the above, knowing that new geometries have been appearing on the market, structured development models for computer mice such as the one proposed here were not found in the literature.
  5.  More information for the four concepts are needed (e.g. Advantage, disadvantage, difference) Answer: The information presented in the revised manuscript satisfies the reviewer's suggestion (5) (#2) (“More information for the four concepts are needed”) , with information on this having been added in section 2.1.2. Concept Generation.
  6.  The innovation and improvement of the final two concepts need to be addressed clearly. Answer: information was added regarding usability (efficiency) and muscular activity the results of which allow to compare the new geometries with commercial devices in an objective way.
  7.  Discussion is too less. To conduct an experiment to evaluate the human usability, physical responses and motions is highly recommended. Comparing the difference, improvement and advantage with the other available products are also needed. The discussion section has completely put the cart before the horse, including the objectives and implications, without seeing a meaningful discussion of the content of this study. Answer: information was added regarding usability (efficiency) and muscular activity the results of which allow to compare the new geometries with commercial devices in an objective way.
  8. The innovation point of the method proposed in this paper to evaluate the shape of pc mice products is not prominent enough. Answer: the purpose of this article is to propose a method to create/develop new geometries having as reference information emanating from regulations, ergonomic principles and related scientific literature; however, information was added regarding the evaluation and validation of the new geometries, the content of which was intended to be used in future publications so as not to make this article too long and “heavy”.
  9.  The introduction of the shape of mouse series products is not detailed. Answer: the authors don't quite understand the reviewer's comment, however information was added in section 2.1.2. Concept Generation.
  10. Figure1: ergonomics evaluation section is questionable. Firstly, please explain why the article uses usability and muscle activity, and their reliability. Secondly, the quantitative study of their factors is needed, which is what this paper lacks. Answer: information was added regarding the evaluation and validation of the new geometries regarding usability and muscle activity, the content of which had been intended to be used in future publications so as not to make this article too long and “heavy”.
  11. The results section in lines 252-264 is incomprehensible and does not provide a good analysis and description of the data from table 2, as well as the force and dimension measurements. Answer: The results section has now rewritten taking advantage of the added information regarding usability (efficiency) and EMG.
  12. 35 and 36 lines said that computer users now use the mouse three times more than the keyboard, but this data is completely unsourced. Answer: reference [3]

Reviewer 3 Report

Very good paper, The authors asked themselves a question about the ergonomics of computer mice. Many users complain of the pain associated with prolonged use of the mouse because we need to keep the hand horizontally. Meanwhile, the natural position of the hand on the table is slightly turned. The proposed new form of the mouse visually meets the condition of ergonomics - minimal energy expenditure on maintaining the position of the hand on the mouse. I propose to test for computer players where fast response is required.

Author Response

Very good paper, The authors asked themselves a question about the ergonomics of computer mice. Many users complain of the pain associated with prolonged use of the mouse because we need to keep the hand horizontally. Meanwhile, the natural position of the hand on the table is slightly turned. The proposed new form of the mouse visually meets the condition of ergonomics - minimal energy expenditure on maintaining the position of the hand on the mouse. I propose to test for computer players where fast response is required.

Answer: included in future works (the discussion and conclusion have been enlarged).

Round 2

Reviewer 1 Report

All the recommendations made were included in the new version. Now the work presents greater scientific solidity, especially in the demonstration of the results achieved.

Author Response

All the recommendations made were included in the new version. Now the work presents greater scientific solidity, especially in the demonstration of the results achieved.

Answer: Thank you! We have extended the discussion and the conclusion further for added clarity and potentially enhanced impact.

Reviewer 2 Report

Thank you very much for the major revision made by the author. The quality of the manuscript has been improved.

Although the usability experiment has been conducted and enriched the content in this revision, the comparison and discussion between the final recommended mouse and the existing mouse is still too less.

Additionally, all the design of the selected mouse types are already common in the market. Hence, the contributions of the article are relatively lacking.

It is recommended to increase the analysis of the pros and cons of each product and to conduct a more in-depth discussion with the results of past related studies. For example: why a certain design (angles, slope...) has a better user experience.

Author Response

Thank you very much for the major revision made by the author. The quality of the manuscript has been improved.

Although the usability experiment has been conducted and enriched the content in this revision, the comparison and discussion between the final recommended mouse and the existing mouse is still too less.

Additionally, all the design of the selected mouse types are already common in the market. Hence, the contributions of the article are relatively lacking.

It is recommended to increase the analysis of the pros and cons of each product and to conduct a more in-depth discussion with the results of past related studies. For example: why a certain design (angles, slope...) has a better user experience.

Answer:
Thank you for your recommendations which  guide us in improving the readability, quality and impact of the manuscript. We have enhanced the discussion and conclusion sections in accordance with your recommendation, as follows:

Table 4 presents the salient characteristics, from a favorable and unfavorable viewpoint, for each one of the five PC mice geometries tested for usability and muscular activation (monitored through surface electromyography of four forearm muscles).

Table 4 – Summary of relevant dichotomous highlights of the tested PC mice geometries

Geometries (slant angle)

Favourable salient aspects

Unfavourable salient aspects

ev 90 deg

Neutral forearm posture (balanced between supination-pronation)

Lowest pointing and clicking efficiency (highest rate of errors in inexperienced use)

ak 60 deg

Best compromise between usability tests and electromyographic analysis performed

Shark fin geometry – not inclusive (restrictive hand sizes)

ci* 45 deg

APDF50 APL best of the tests (pointing large)

Buttons are hidden by the body of the PC mouse (medium error rate in inexperienced use)

ch* 30 deg

High pointing and clicking efficiency (low rate of errors in inexperienced use); thumb support included in the geometry

Low pointing and clicking efficiency (high rate of errors in inexperienced use)

mi 0 deg

Highest pointing and clicking efficiency (lowest rate of errors in inexperienced use)

Full pronation of the forearm  

* - These PC mice geometries (ci and ch) were designed and prototyped by the first author in 2017 for his PhD thesis; at that time, in addition to not being possible to identify from scientific literature, identic geometries, there were no commercially available models embodying these PC mice geometries.

Usability results are comparable to the benchmark models, in particular for the slanted and vertical models (ak and ev). Surface electromyographic study for the pointing task shows a significantly decreased APL activation for the ci model at APDF50, pointing to the interest in industrializing and marketing this geometry for enhanced computer mouse ergonomics. Future studies should consider testing the new geometries using an array of other kinds of tasks as well as including gaming applications in the tests, and the test by computer games, where added speed is valued in addition to ease of use and comfortable handling.

The current results are aligned with previous work by Odell and Johnson [3] recommending increasing mouse height and angling the mouse can improve wrist posture, without negatively affecting performance; this is particularly notorious in prototype ci, but also noticeable for prototype ch. Moreover, the current study also corroborates previous work by Agarabi et al. [7] showing a significant decrease in the level of sEMG activity for selected muscles when subjects were tested using ergonomic computer mice, which in the current study was applicable to APL in pointing large for APDF50, with regard to the ci geometry. Similar to previous results, of a study investigating use of mice allowing more neutral posture of the wrist [18], the current study shows that the ci prototype mouse falls well within the range of performance measures associated with already existing commercially available input devices.

[... in the conclusion section]

This study contributes to enhance the knowledge related to PC usage, specially related to computer pointing devices development, with focused contributions in the following aspects:

  1. compilation of requirements emanating from regulations and other related literature for specification of new handheld computer pointing devices,
  2. method for developing and selecting (choosing) innovative PC mice geometries following previously defined criteria, and
  3. evaluation tools applicable to the products developed (usability, muscle activity).